# Removing unwanted variation with CytofRUV to integrate multiple CyTOF datasets

**Marie Trussart[1,2]\*, Charis E Teh[3,4], Tania Tan[3,4], Lawrence Leong[1,2], Daniel HD Gray[3,4], Terence P Speed[1,2]**

[1]Bioinformatics Division, Walter and Eliza Hall Institute of Medical Research, Parkville, Australia; [2]School of Mathematics and Statistics, The University of Melbourne, Melbourne, Australia; [3]The Walter and Eliza Hall Institute of Medical Research, Parkville, Australia; [4]Department of Medical Biology, The University of Melbourne, Parkville, Australia

**Abstract** Mass cytometry (CyTOF) is a technology that has revolutionised single-cell biology. By detecting over 40 proteins on millions of single cells, CyTOF allows the characterisation of cell subpopulations in unprecedented detail. However, most CyTOF studies require the integration of data from multiple CyTOF batches usually acquired on different days and possibly at different sites. To date, the integration of CyTOF datasets remains a challenge due to technical differences arising in multiple batches. To overcome this limitation, we developed an approach called CytofRUV for analysing multiple CyTOF batches, which includes an R-Shiny application with diagnostic plots. CytofRUV can correct for batch effects and integrate data from large numbers of patients and conditions across batches, to confidently compare cellular changes and correlate these with clinically relevant outcomes.

**\*For correspondence:**
trussart.m@wehi.edu.au

**Competing interests:** The authors declare that no competing interests exist.

## Introduction

Mass cytometry or Cytometry by Time-Of-Flight (CyTOF) (*Bandura et al., 2009*) is a high-throughput technology that permits the simultaneous measurement of the expression level of more than 40 proteins in millions of single cells. It uses antibodies, which are labelled with heavy metal ion tags to target the proteins of interest, and are in turn detected by time-of-flight mass spectrometry. CyTOF has been a powerful tool for delineating cell subsets in heterogeneous tissues such as blood and tumour, and for correlating single-cell differences with biologically relevant outcomes (*Levine et al., 2015*; *Qiu et al., 2011*). This capability has been useful in understanding the mechanisms of resistance that develop in certain blood cancers to a new class of anti-cancer drugs, termed BH3 mimetics, in early stage clinical trials of several blood cancers (*Agarwal et al., 2019*; *Blombery et al., 2019*). Yet, a major challenge in the field is the high variation in the performance observed in the CyTOF instrument, caused by both differences in instrument calibration and fluctuations in signal strength.

To overcome this challenge, a normalisation method was created to improve the comparability between measurements (*Finck et al., 2013*). Briefly, the method involves the addition of five types of control beads mixed in with cells, each type tagged with a different heavy metal element, the collection of the control beads throughout the run, and the application of a multiplicative correction at the end of the run. For the multiplicative correction, the algorithm calculates smoothed intensities from each control bead element, estimates a coefficient at each control bead acquisition time-point, and corrects the instrument sensitivity at that specific time-point by computing a unique slope for all the control bead elements, assuming that they vary at similar rates. To extend from control bead

events to all cells, the value of the coefficient is linearly interpolated over all time-points of the experiment by assuming that all cells (non-control bead events) have similar slopes to the closest bead time-point. The final normalisation step involves applying the interpolated correction coefficient to all the protein measurements. The use of this normalisation to account for intra-instrument time-drift variation has become common practice, but novel correction procedures are still needed to address all other types of variation between samples between research centres, see for example (*Leipold et al., 2018*) and within laboratories run on different days (see below).

In this study, a single CyTOF dataset, barcoded sample set, or run is referred to as a CyTOF batch. We call the batch effects we seek to remove 'unwanted variation', and their causes include: differential antibody staining across samples within a batch, different batches of reagents, different machines or the inevitable lab differences found in multicentre studies. The ability to accurately distinguish true biological changes from technical artefacts like those just mentioned is critical, and has already been done to an extent for flow cytometry assays (*Finak et al., 2016*; *Maecker et al., 2012*).

Two methods have recently been published that aim to achieve consistency between samples across batches which make use of shared reference samples across batches. BatchAdjust (*Schuyler et al., 2019*) offers methods analogous to the control bead normalisation described above, which include scaling all measurements by ratios of means or medians. With the scaling methods, a factor is computed for each protein and each batch to adjust the measurements on the reference samples replicated across batches. Similar adjustments are then applied to the samples within a batch to achieve consistency with their reference samples. However, technical variation can impact specific cell types differently (*Van Gassen et al., 2019*). To address this, CytoNorm (*Van Gassen et al., 2019*) uses the clustering algorithm FlowSOM (*Van Gassen et al., 2015*) to identify clusters prior to normalisation, and defines a cluster-specific goal distribution for the values of each protein measurement using the means of quantiles. Their approach then uses splines to transform the original protein values into new values, which have the goal distribution. This method relies on the strong assumption that batch effects do not affect the clusters, and this is examined using coefficients of variation. Both methods were shown to be effective in removing batch-to-batch variation in the datasets analysed. The observation that batch differences can affect clusters differently (*Van Gassen et al., 2019*) suggests that it will not be sufficient to apply a single batch adjustment to the measurements on all cells, as it is being done in BatchAdjust (*Schuyler et al., 2019*). However, a comparison of the two methods assessing the performance and limitations of each method has not been performed. Additionally, there are no tools or metrics to assess whether the post normalised CyTOF data are more or less consistent across batches, not only at the protein expression level but also at the cluster level.

In recent years, a class of methods called Remove Unwanted Variation (RUV) has been developed to remove unwanted variation such as batch effects, from high-dimensional genetic and genomic data. They have been applied to microarray (*Gagnon-Bartsch and Speed, 2012*), RNA-seq (*Risso et al., 2014*), Nanostring nCounter gene expression (*Molania et al., 2019*) and single-cell RNA-seq data (*Lin et al., 2019*). Here, for the first time, we adopt the approach and develop a computational algorithm which permits the integration of data across CyTOF batches. Our method is based on the RUV-III method (*Molania et al., 2019*), which uses technical replicates and negative control genes to estimate unwanted variation. We applied RUV-III to CyTOF data by exploiting pseudo-replicates to estimate the unwanted variation and remove it. It is implemented in the R package 'CytofRUV', which is available at the following link: www.github.com/mtrussart/CytofRUV. We begin by examining the batch effects found when comparing CyTOF data from samples replicated across batches. To do so, we built an R-Shiny application that exhibits any batch effects present in such samples using four different diagnostic plots and their associated numerical metrics based on protein expression distributions and clustering results. Then, we compare the unadjusted data with the normalised data using either CytofRUV, BatchAdjust or CytoNorm, all on three different datasets. Our results suggest that not only does CytofRUV do better at removing unwanted variation from measured protein expression, it also makes the distributions of these quantities more uniform across batches, and enhances the detection of biologically important changes embodied in the data across batches.

## Results

### Batch effects include protein expression differences

A known source of unwanted variation in CyTOF datasets is the time-drift in signal intensity (*Finck et al., 2013*). However, there can also be variation due to differences across batches in antibody conjugates or other reagents, as well as operators, machines and laboratories. To exhibit some of the batch effects that can arise with CyTOF within one lab, we conducted an experiment using samples replicated across batches. Replicated samples allow us to assess intra-site reproducibility and systematic differences due to technical variation.

We created a dataset based on 24 samples in total, consisting of peripheral blood mononuclear cell (PBMC) samples from three patients with chronic lymphocytic leukaemia (CLL) and PBMC from nine healthy controls (HC), each replicated across two batches of 12 samples (*Table 1*). All samples were stained with a 31-antibody panel targeting 19 lineage (*Table 2*) and 12 functional proteins (*Table 3*) that were previously validated (*Teh et al., 2020*). After processing the data (Methods), we applied an arcsinh-transformation defined as *arcsinh (intensity/5)* in all that follows.

The first class of diagnostic plots we use is based on median protein expression. The multi-dimensional scaling (MDS) plot (*Figure 1A*) computed using median protein expression from all cells in each sample as described in *Crowell HL et al., 2017* and *Nowicka et al., 2017*, shows the dissimilarities between samples. The first dimension (MDS1) separates the CLL from the HC samples well.

**Table 1.** Samples descriptions.
The first column indicates the sample id, the second the patient condition, either healthy controls (HC) or chronic lymphocytic leukaemia (CLL), the third column indicates the patient id and the last indicates the batch number, 1 or 2.

| Sample Id | Condition | Patient Id | Batch |
| --- | --- | --- | --- |
| HC1_B1 | HC | VBDR996 | 1 |
| HC2_B1 | HC | VBDR1089 | 1 |
| HC3_B1 | HC | VBDR1090 | 1 |
| HC4_B1 | HC | VDBR1098 | 1 |
| HC5_B1 | HC | VDBR1108 | 1 |
| HC6_B1 | HC | VDBR1103 | 1 |
| HC7_B1 | HC | VDBR1105 | 1 |
| HC8_B1 | HC | VDBR1107 | 1 |
| HC9_B1 | HC | VBDR1111 | 1 |
| CLL1_B1 | CLL | DG33-01 | 1 |
| CLL2_B1 | CLL | DG23-01 | 1 |
| CLL3_B1 | CLL | DG27-01 | 1 |
| HC1_B2 | HC | VBDR996 | 2 |
| HC2_B2 | HC | VBDR1089 | 2 |
| HC3_B2 | HC | VBDR1090 | 2 |
| HC4_B2 | HC | VDBR1098 | 2 |
| HC5_B2 | HC | VDBR1108 | 2 |
| HC6_B2 | HC | VDBR1103 | 2 |
| HC7_B2 | HC | VDBR1105 | 2 |
| HC8_B2 | HC | VDBR1107 | 2 |
| HC9_B2 | HC | VBDR1111 | 2 |
| CLL1_B2 | CLL | DG33-01 | 2 |
| CLL2_B2 | CLL | DG23-01 | 2 |
| CLL3_B2 | CLL | DG27-01 | 2 |

**Table 2.** Lineage surface proteins selected.

The first column indicates the transition element isotope (mass number, element name), the second column indicates the antigen selected, and the last two columns indicate the clone name and vendor.

|  | Metal | Lineage (surface) protein antibody | Clone | Vendor |
|---|---|---|---|---|
| 1 | 89 Y | CD45 | HI30 | BioLegend |
| 2 | 115 In | HLA-DR | L243 | BioLegend |
| 3 | 140 Ce | CD27 | M-T271 | BioLegend |
| 4 | 141 Pr | CD235a/b | HIR2 | BioLegend |
| 5 | 142 Nd | CD19 | HIB19 | BioLegend |
| 6 | 143 Nd | CD5 | UCHT2 | BioLegend |
| 7 | 144 Nd | CD38 | HIT2 | BioLegend |
| 8 | 145 Nd | CD4 | RPA-T4 | BioLegend |
| 9 | 146 Nd | CD8 | RPA-T8 | BioLegend |
| 10 | 147 Sm | CD20 | H1 | BD |
| 11 | 148 Nd | CD16 | 3G8 | BioLegend |
| 12 | 151 Eu | CD123 | 6H6 | BioLegend |
| 13 | 155 Gd | CD56 | B159 | BioLegend |
| 14 | 156 Gd | CD14 | HCD56 | BioLegend |
| 15 | 159 Tb | CD11c | Bu15 | BioLegend |
| 16 | 169 Tm | CD45RA | HI100 | BioLegend |
| 17 | 170 Er | CD3 | UCHT1 | BioLegend |
| 18 | 171 Yb | CD66 | CD66a-B1.1 | DVS |
| 19 | 209 Bi | CD61 | VI-PL2 | DVS |

The second dimension (MDS2) shows the batch differences, with samples that originate from batch one placed at the bottom of the plot and samples from batch two at the top of the plot. This distinction clearly reveals that the protein expression measured in the samples is affected by batch. We also carried out hierarchical clustering on the median expression across all cells in the samples of the 19 lineage proteins and 12 functional proteins detected, to highlight the proteins driving the

**Table 3.** Set of intracellular functional proteins selected.

The first column transition element isotope (mass number, element name), the second column indicates the antigen selected, and the last two columns indicate the clone name and vendor.

|  | Metal | Functional (intracellular) protein antibody | Clone | Vendor |
|---|---|---|---|---|
| 1 | 140 Ce | BAK | 7D10 | WEHI |
| 2 | 153 Eu | Bcl-xL | E18 | Abcam |
| 3 | 154 Sm | Bax | 1B4 | WEHI |
| 4 | 157 Gd | Bcl-2 | 100 | WEHI |
| 5 | 160 Gd | Mcl-1 | Y37 | Abcam |
| 6 | 161 Dy | cMyc | D84C12 | CST |
| 7 | 163 Dy | BFL-1 | SP435 | Abcam |
| 8 | 165 Ho | Bim | 3C5 | WEHI |
| 9 | 166 Er | pRb [S807/811] | J112-906 | BD |
| 10 | 172 Yb | BCLW | 16H12 | WEHI |
| 11 | 173 Yb | cCaspase3 | C92-605 | BD |
| 12 | 174 Yb | p53 | 7F5 | CST |

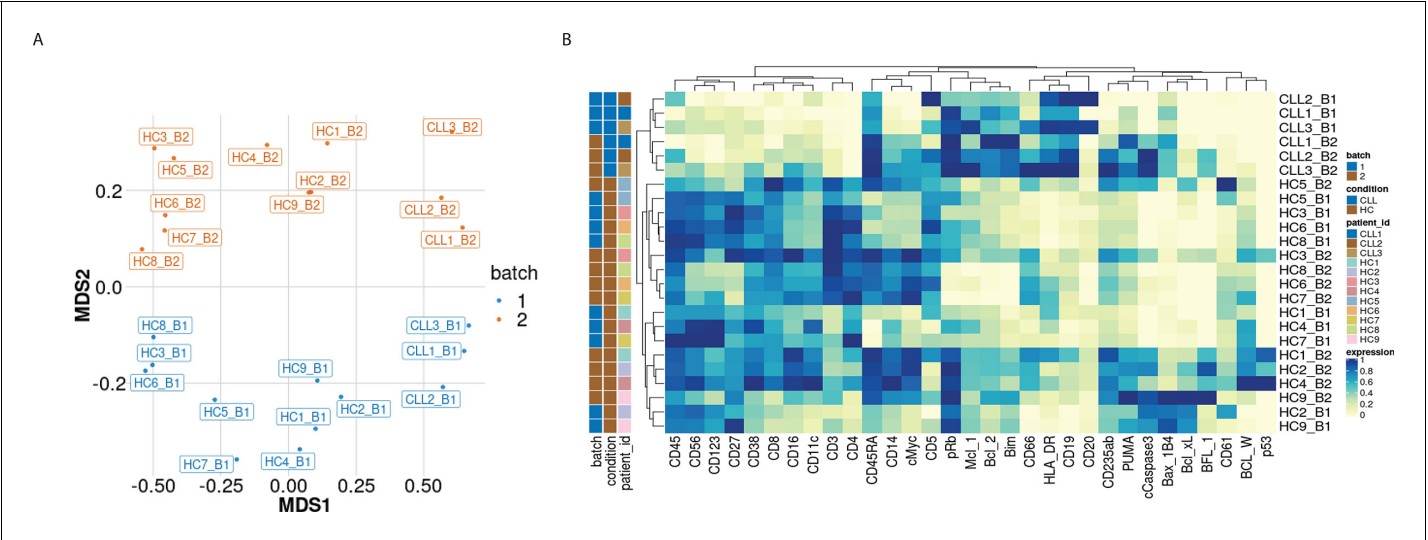

**Figure 1.** Visualisation of batch effects on the median protein expression across batches . (**A**) Multi-dimensional scaling plot of the 24 samples computed using median protein expression. (**B**) Heatmap of the median protein expression of 19 lineage proteins and 12 functional proteins across all cells measured for each sample in the dataset.

observed clustering of samples (columns) and proteins (rows) in the heatmap (*Figure 1B*). As with the MDS plot, a grouping of samples by condition and by batch is observed.

We next examined the magnitude of the batch-to-batch differences in the distributions of the protein expression across replicates, as our second class of diagnostic plots. We observed that batch effects not only affect each protein differently (*Figure 2—figure supplement 1*) but also each sample differently (*Figure 2—figure supplement 1*). To assess the importance of the variation found between samples replicated across batches, we compared them to expected, biologically relevant differences within a single batch. To do this, we used another dataset of a replicated sample from one index patient with CLL, and samples from six other patients with CLL. The one sample from the index patient with CLL was replicated across eight different CyTOF batches, resulting in eight replicated CLL datasets. We compared the variation in these datasets with that from six other patients with CLL processed in a single CyTOF batch. We focussed on expression of BCL-2, an archetypal pro-survival protein that is greatly upregulated in CLL cells compared to their normal B cell counterparts, yet still exhibits variation in CLL cells among patients (*Majid et al., 2008*). We found that the variation in the distributions of the BCL-2 expression from a single sample across batches (*Figure 2A*) is comparable to that observed in the BCL-2 expression for the six patients at screening from a single CyTOF batch (*Figure 2B*). We conclude that batch effects due to unwanted variation can occur over a range comparable to that due to actual biological changes among patients (*Figure 2A–B*). An important repercussion in this example is that the impacts of treatments on distinct CLL cellular phenotypes would not be confidently detected without correcting for batch effects.

## Batch effects affect cell subpopulations differently

To assess how batch effects affect cell subpopulations, we used our first dataset of samples from 3 patients with CLL and 9 PBMC healthy controls (HC). We partitioned the entire collection of cells into clusters using FlowSOM as described in *Nowicka et al., 2017*, which appeared among the fastest and best performing clustering approaches for CyTOF data (*Weber and Robinson, 2016*). In all 24 samples, we carried out this clustering using the 19 lineage proteins to identify 20 cell clusters (*Figure 3A* and *Figure 3—figure supplement 1*). We performed the clustering using different numbers of clusters, and the choice of 20 clusters was determined based on the biological interpretation of the cell subpopulations found. Here we refer to clusters as cell subpopulations, although in some cases the clustering method might produce clusters that do not necessarily correspond to homogeneous cell subpopulations. The third class of diagnostic plots we display are based on the clustering

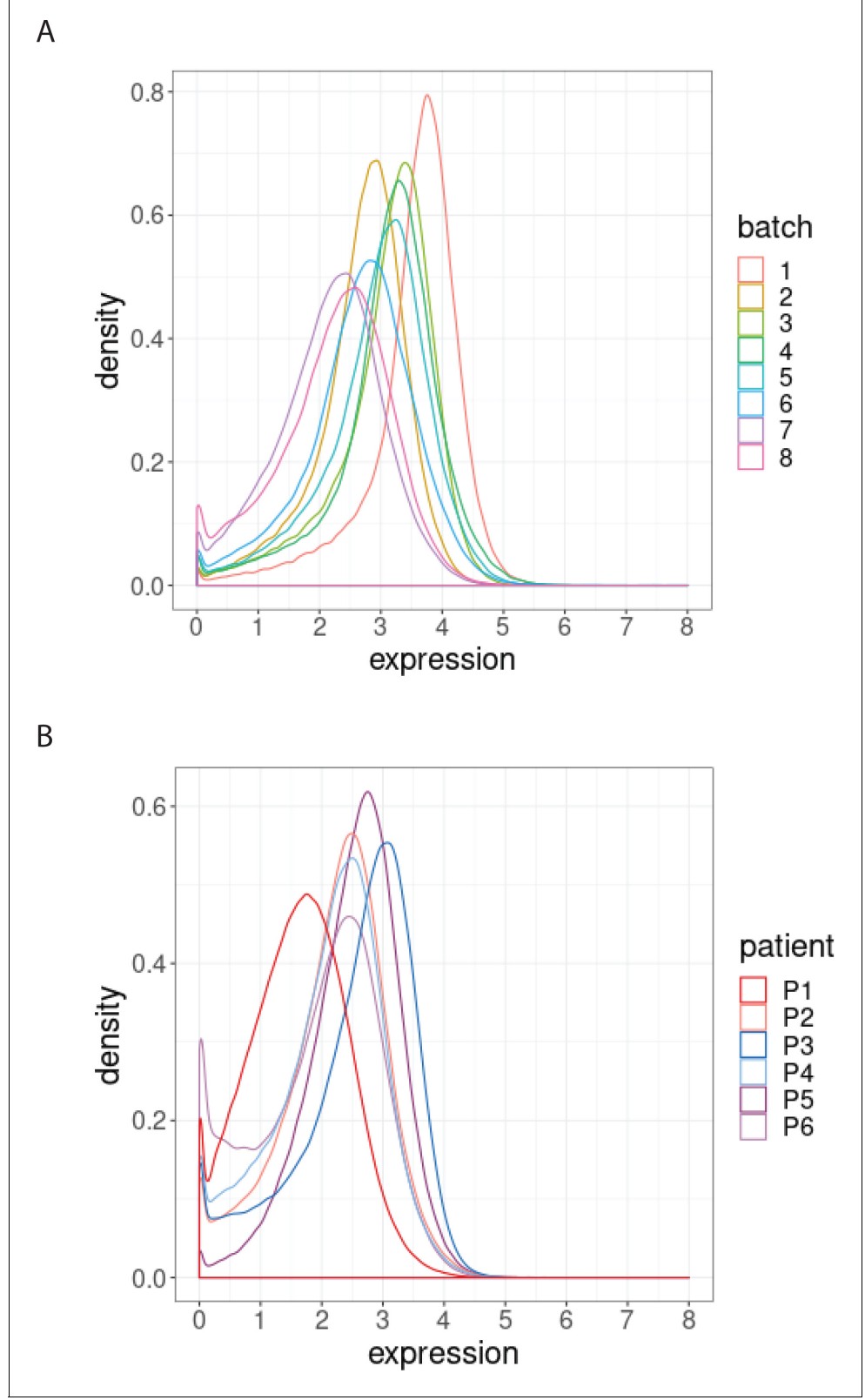

**Figure 2.** Distribution of BCL-2 expression. (A) Distributions of BCL-2 expression in one sample from one treated CLL cancer patient, replicated across 8 CyTOF batches, coloured by batch. (B) Distributions of BCL-2 expression in one sample from each of 6 different CLL cancer patients at screening, processed in a single CyTOF batch, coloured by patient.

*Figure 2 continued on next page*

*Figure 2 continued*

The online version of this article includes the following figure supplement(s) for figure 2:

**Figure supplement 1.** Protein distributions before normalisation for the samples CLL2 and HC1 across batch 1 and batch 2.

**Figure supplement 2.** Protein distributions after CytofRUV normalisation with k = 5 for the samples CLL2 and HC1 across batch 1 and batch 2.

results. We used t-Distributed Stochastic Neighbor Embedding (t-SNE) as described in *Crowell HL et al., 2017* and *Nowicka et al., 2017* to give a 2D representation of the single-cell data, with the positions of cells reflecting their proximity in high-dimensional space. It was then possible to visualise the impact of batch differences on cell subtypes identification in the datasets by examining the t-SNE plot coloured by cluster (*Figure 3A*), which assumed different distributions across the replicate CyTOF runs. Additionally, we highlighted the batch effect by overlaying the predominant CLL cell subpopulation (cluster 9) coloured by batch (*Figure 3B*). The distinct positions of these clusters on the second dimension (t-SNE2) suggested that substantial unwanted variation was altering cell population measures across the two batches.

To optimise the dimensionality reduction and visualise the extent to which discrete subsets of cells are separated from each other in the 31-dimensional space, we performed a linear discriminant

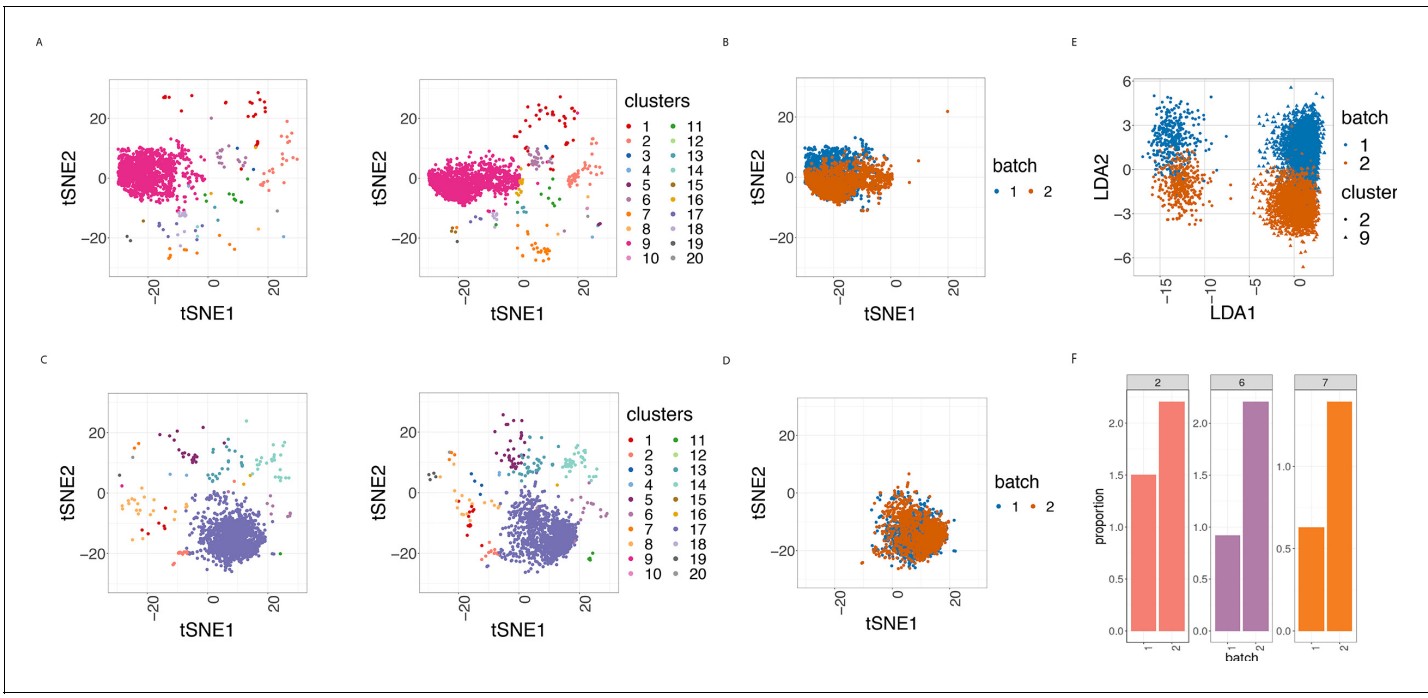

**Figure 3.** Cell clustering plots show batch effects in cells from the same cancer patient CLL1 sample replicated across 2 CyTOF runs. (A) Cell clustering identification. t-SNE plot based on the arcsinh-transformed expression of the 19 lineage proteins in the cells. For display purposes, 2000 cells were randomly selected from each of the samples. Cells are coloured according to the 20 clusters obtained using FlowSOM clustering stratified by batch 1 (left) or 2 (right) of the corresponding replicated sample. (B) Same as in (A) selecting only cluster 9 cells but coloured by the batch 1 or 2 of the corresponding replicated sample. (C) Same as in (A) but after CytofRUV normalisation with k = 5. (D) Same as in (B) but after CytofRUV normalisation with k = 5. (E) Linear discriminant analysis applied to data on two cell types from the same sample replicated across two batches, with shape indicating cell type and colour indicating batch. (F) Cluster proportions. Barplot of the relative abundance (percentage) of the cells in clusters 2, 6 and 7 by batch. The online version of this article includes the following figure supplement(s) for figure 3:

**Figure supplement 1.** Heatmap of the median lineage protein expression across clusters with the associated cluster percentages measured for all cells and all samples in the first dataset of samples from 3 patients with CLL and 9 HC.

**Figure supplement 2.** BCL-2 median expression in the main CLL cluster from the 3 CLL samples replicated across 2 CyTOF runs.

**Figure supplement 3.** Linear discriminant analysis plot to show batch effects in cells from the same cancer patient CLL1 sample replicated across batches after CytofRUV normalisation with k = 5.

**Figure supplement 4.** Boxplot of the differences of median protein expression differences across batches before and after CytofRUV normalisation (ΔΔ, see Materials and methods) with k=5 within the main prominent cell subpopulation.

analysis (LDA) on four subsets of cells: the predominant CLL cell subpopulation (cluster 9) and CD8 killer cell subpopulation (cluster 2) identified in the study from batches 1 and batch 2. The first dimension (LD1) separates the cell types well, while the second dimension (LD2) embodies batch differences: cells that originate from batch one are located at the top of the plot while cells from batch two are at the bottom of the plot (*Figure 3E*).

CyTOF has the unique ability to provide a deeper understanding of the molecular changes induced by targeted therapies at the single-cell level and how their efficacy is influenced by cancer heterogeneity. However, the assessment of patient heterogeneity and its correlation with the clinical outcome necessitates a reliable detection of changes in protein expression across patients. As such, we compared the median expression of BCL-2 for the 3 patients with CCL in the main CLL cluster nine and found that the variation from a single sample across batches is higher than that observed in the BCL-2 median expression across patients (*Figure 3—figure supplement 2A*). To overcome this limitation, the assessment of heterogeneity of patients with CLL requires a correction for batch effects that would remove this unwanted variation.

## Batch effects can induce differences in cell subpopulation abundances

In CyTOF studies, the analysis of cell subpopulation abundances as well as that of protein expression can be used to identify sets of proteins that are associated with response to a treatment. Comparing the proportions of inferred cell types across different drug treatments highlights the subpopulations that change across experimental conditions. To assess whether batch differences affect cell subpopulation abundances, we compared the cluster frequencies across replicates. We detected a noticeable difference in the proportions of CLL cancer cells (cluster 2, cluster 6 and cluster 7) among cells that originate from the batch one compared to those from batch 2 (*Figure 3F*). Our R-Shiny application can also be used to visualise the cluster proportions across samples, and is our fourth class of diagnostic plots (*Figure 4*). Such variation in cell subpopulation abundance is important when batches have markedly different proportions. We need to be able to identify changes in subpopulation abundance which are due to biology and not to unwanted variation between samples.

In summary, we examined the reproducibility of samples replicated across batches. To facilitate the identification of batch effects across replicates, we developed an R-Shiny application that produces the four diagnostic plots previously mentioned: Median Protein Expression, Protein Expression Distributions, Clustering Results and Cluster Proportions. We found that batch differences affect not only the protein expression levels, but also the cluster proportions. Such differences are problematic in large-scale studies with multiple patients, cell types and treatments, as they compromise the detection of biologically important changes. The integration of datasets from multiple CyTOF batches is therefore an important challenge to be addressed.

## Using CytofRUV to remove batch effects in CyTOF data

To integrate data from multiple CyTOF batches, we have developed a normalisation method that removes batch effects in CyTOF data. In order to estimate and adjust for such unwanted variation, CytofRUV exploits the concept of pseudo-replicates (here cells) in the RUV-III method that has been successfully applied to the Nanostring nCounter gene expression platform (*Molania et al., 2019*) and to single-cell RNA-seq data (*Lin et al., 2019*). We cluster using FlowSOM, and we assume that at least one cluster in the replicated samples is shared across the batches. We then consider the cells of clusters shared across the batches to be pseudo-replicates (see Methods). To adjust for batch effects, CytofRUV begins by averaging protein values across pseudo-replicates, and then forms residuals. This leads to an estimate of one aspect of the unwanted variation ($\alpha$) on each protein, which in turn is used to estimate the other aspect (W) of the unwanted variation on each cell. Finally, those estimates are combined into an estimate of the unwanted variation (W$\alpha$), and that is subtracted from the data. The dimension (k) of the unwanted variation also needs to be determined. To find a good value for k, we repeat the analysis with different values of k and then evaluate the quality of each result using our diagnostic plots and the corresponding summary statistics.

We first used CytofRUV on data from the 12 samples replicated across two batches, using two samples as our known replicated reference samples. Specifically, we used HC1 for the HC samples and CLL2 for the CLL samples, defining all cells from those samples in any given cluster to be pseudo-replicates. Assuming that all 20 clusters have cells from HC1 and CLL2 in both batches, any

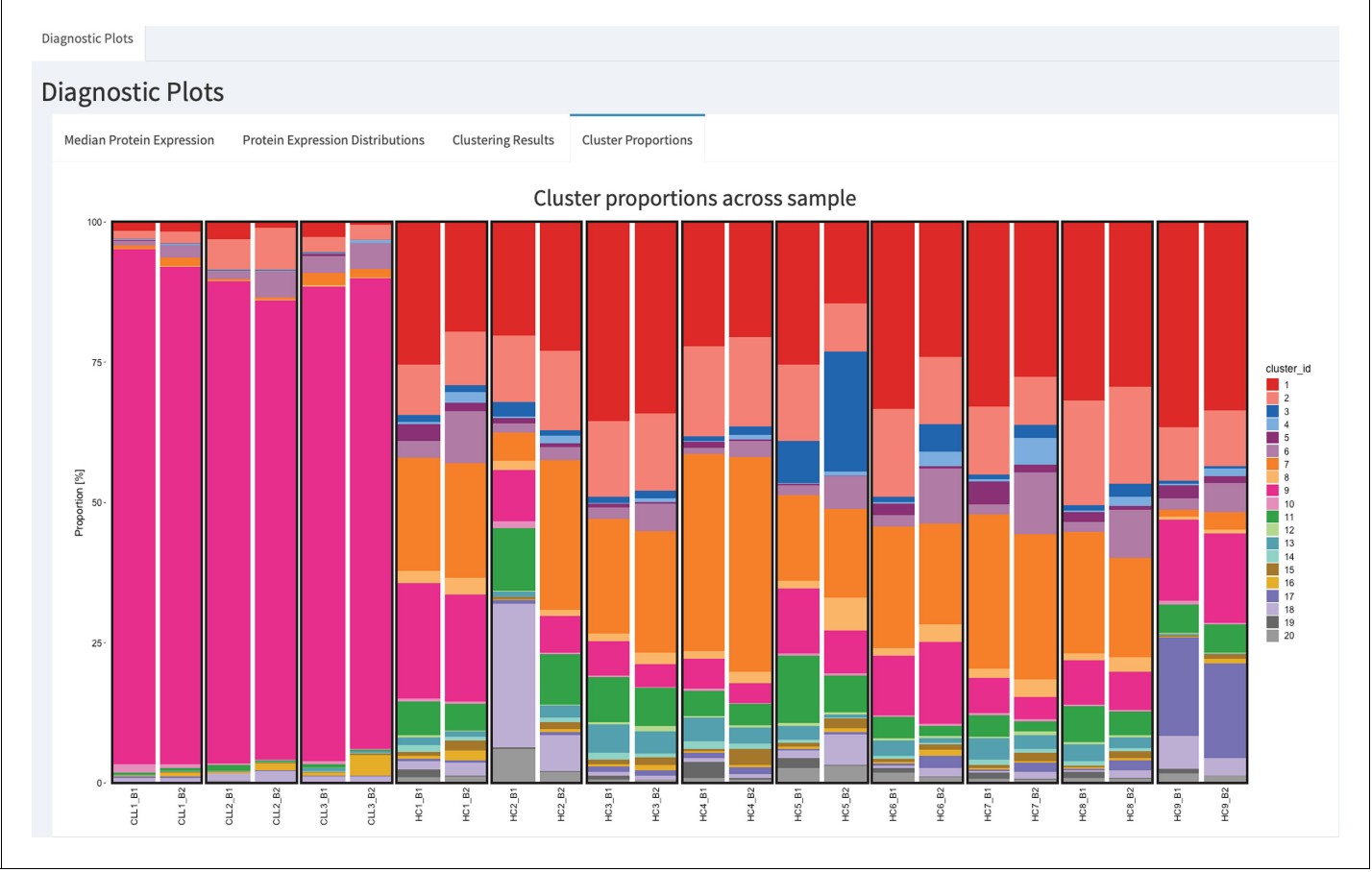

**Figure 4.** CytofRUV's R-Shiny application for the identification of batch effects in cluster proportions across batches. All diagnostic plots can be obtained by the user selecting an option at the top left corner by from: Median Protein Expression, Protein Expression Distributions, Clustering Results and Cluster Proportions. The selected option displays barplots of cluster proportions across samples before normalisation and by conditions CLL or HC on a subsample of the whole dataset. Vertical black boxes contain the same replicated sample across batches one and batch2.

differences in protein expression between cells within the cluster but in different batches will represent unwanted variation. We summarized the performance of our CytofRUV normalisation method on all the CLL samples using three metrics and compare them with the corresponding ones for BatchAdjust (*Schuyler et al., 2019*) and CytoNorm (*Van Gassen et al., 2019*).

## CytofRUV reduces batch effects from protein expression

To assess the quality of the our CytofRUV normalisation, we first compared the distributions of proteins across the two batches for the designated replicated reference samples. We found that, for all the proteins, these distributions become more similar across batches (*Figure 2—figure supplement 1*, *Figure 2—figure supplement 2*). Also, we observed that the variation in the median expression of BCL-2 for the 3 patients with CLL in the main CLL cluster is reduced compared to that observed in the BCL-2 median expression across patients (*Figure 3—figure supplement 2B*). We also observed a decrease in the batch effects both on the t-SNE plots (*Figure 3C and D*) and in the linear discriminant analysis (*Figure 3—figure supplement 3*). To quantify the batch differences between these pairs of distributions, we computed the Earth Movers Distance (EMD) as described in *Van Gassen et al., 2019* for all the proteins and all CLL and HC samples for both the original dataset and the normalised datasets (*Figure 5A and D*). For only the CytofRUV normalisation, we also computed the EMD by cluster for all proteins to assess the reduction in batch differences within cluster compared to the raw data, where batch effects are affecting protein distributions within cluster differently (*Figure 5—figure supplement 1*). For all CLL samples, CytofRUV gave the lowest EMD for all values of k = 5,10 and 15, not only compared to the raw data but also compared to both

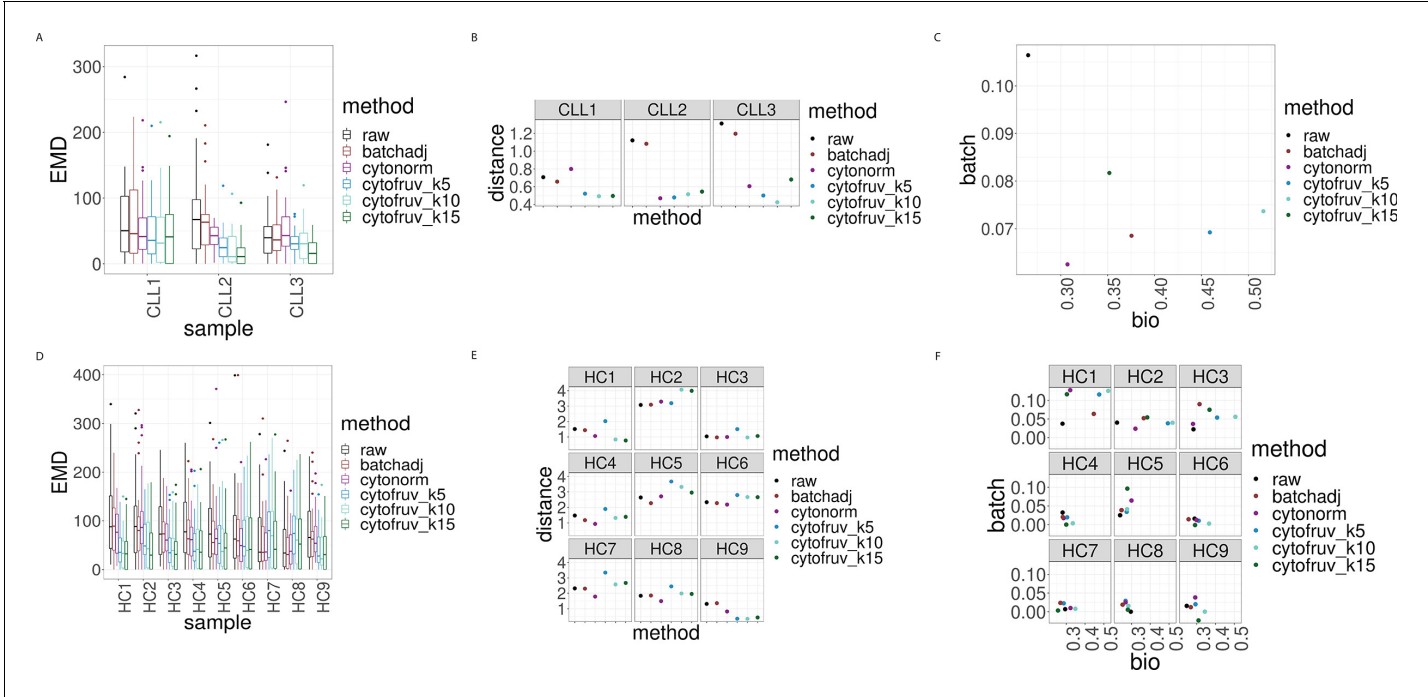

**Figure 5.** Metrics to assess the effectiveness of the normalisation methods. In all panels, the colour indicates either the raw data or the method used for normalisation. (A) Boxplots of the Earth Movers Distances (EMD) between paired protein expression distributions across batches for each CLL sample. (B) Hellinger distances between paired cluster proportions across batches for each CLL sample. (C) Mean Silhouette scores computed for all CLL samples on the cluster types (bio) on the x-axis and on batch (batch) on the y-axis. (D) Same as (A) for the HC samples. (E) Same as (B) for the HC samples. (F) Silhouette scores computed for all HC samples on the cluster types (bio) on the x-axis and on batch (batch) on the y-axis.
The online version of this article includes the following figure supplement(s) for figure 5:

**Figure supplement 1.** EMD for all the proteins by cluster, before and after CytofRUV normalisation of the CLL2 sample, k = 5.

BatchAdjust and CytoNorm (*Figure 5A*). Similarly, CytofRUV also gave the lowest EMD for 7 out of 9 HC samples compared to both BatchAdjust and CytoNorm (*Figure 5D*).

Altogether EMD revealed that CytofRUV is the method with the optimal reduction in batch differences from protein expression for 11 out of 12 samples. CytoNorm gave the lower EMD when compared with BatchAdjust for 7 out of 12 samples (*Figure 5A and D*).

## CytofRUV makes the cell subpopulation proportions more consistent across batches

The proportions of the different cell subpopulations in replicates in different batches should be consistent. This consistency among the replicates ensures that a differential analysis of their abundances will confidently detect robust cellular changes across experimental conditions. To assess how well CytofRUV corrects for differences in cluster proportions across batches, we computed the Hellinger distance (see Methods) between the paired cluster proportions of all replicated samples. For the HC samples, CytofNorm gave the lowest Hellinger distances for 4 samples out of 9 compared to both CytofRUV and BatchAdjust. Two remaining samples are the ones where CytofRUV is able to make those proportions more consistent across replicates (*Figure 5E*). For all the CLL samples, the lowest Hellinger distances are found for the normalisation with CytofRUV methods (*Figure 5B*). Overall CytofRUV is able to adjust the cluster proportions and make them consistent across replicates to a greater extent than is achieved by the two others methods. CytoNorm also performs well in adjusting those proportions, while BatchAjust generally leads to Hellinger distances similar to those in the raw data (*Figure 5B and E*).

## Effective removal of unwanted variation leads to a better separation of cell subpopulations

To evaluate whether CytofRUV not only removes batch effects but preserves biology, we compare the cell-to-cell variation within clusters in relation to between clusters. We do this by computing *Silhouette* scores (see Methods), which are a combined measure of the degree of separation of cells within clusters and that between clusters. We compute two Silhouette scores $s_{biology}$ and $s_{batch}$, based on the cell subpopulations defined by the clusters, and batch grouping, respectively. A normalisation method that successfully removes batch effects should lead to adjusted data with a small $s_{batch}$ score, while one which preserves or enhances the biology should have high value of $s_{biology}$, at least as high as that before adjustment. To assess this aspect of the performance of normalisation methods, we combine the $s_{biology}$ for clusters with $s_{batch}$ for the batches in a single plot (*Figure 5C and F*). When the raw data are found at the bottom left corner of the plot and a normalisation method is found at the top right corner of the plot, this indicates that the method has successfully removed batch effects and preserved or enhanced the biology. According to the mean Silhouette scores of the CLL samples (*Figure 5C*), CytofRUV with k = 5 gives the best results, not only enhancing the biology i.e. cell sub-type definition across replicates (also on t-SNE plots *Figure 3C and D*), but also removing the batch effects. For most of the HC samples, $s_{batch}$ is similar for all methods and in a few samples $s_{batch}$ is higher than in the raw data for all methods (*Figure 5F*). Using CytofRUV with k = 5 for the HC samples, we also obtained a higher $s_{biology}$ for 8 out of the 9 HC samples (*Figure 5F*) compared to the two other methods.

## CytofRUV removes batch effects across multiple batches in two other datasets

To expand our analysis, we also tested all methods on two other datasets containing replicate reference samples across multiple batches, taken from those used in *Schuyler et al., 2019* and in CytoNorm (*Van Gassen et al., 2019*).

The first dataset (*Schuyler et al., 2019*) is from peripheral whole blood samples from a single healthy donor, processed at one time to include unstimulated and stimulated conditions (Lipopolysaccharide LPS+ and Resiquimod R848-), and aliquoted into 12 batches. We carried out the normalisation of all samples using BatchAdjust as explained in *Schuyler et al., 2019*. For CytofRUV and CytoNorm, we used all the stimulated samples as replicated reference samples and identified 20 clusters with FlowSOM. CytofRUV used all clusters to define pseudo-replicates.

Our R-Shiny application can also be used to assess the ability of a normalisation method to correct for batch effects. Using our fourth diagnostic plot, we can visualise the ability of CytofRUV with k = 10 to remove batch effects in cluster proportions across 12 batches (*Figure 6C*) compared to that in the raw data (*Figure 6A*). We also summarized the performance of all three methods using the same three metrics (*Figure 6—figure supplement 1*). Again, we observed the ability of CytoNorm to correct for differences in cluster proportions in stimulated samples where the lowest Hellinger distances are found (*Figure 6—figure supplement 1B*), and in unstimulated samples (*Figure 6—figure supplement 1E*), followed by CytofRUV with overall lower distances than BatchAdjust. Likewise, we confirmed that, overall, CytofRUV gave lower EMD for both the stimulated (*Figure 6—figure supplement 1A*) and unstimulated samples (*Figure 6—figure supplement 1D*) compared to the two other methods, which in some cases have some cases higher EMD than those of the raw data. Again we observed improved Silhouette scores for CytofRUV compared with BatchAdjust and CytoNorm, which have higher $s_{batch}$ (*Figure 6—figure supplement 1C*).

The second dataset (*Van Gassen et al., 2019*) comes from the FlowRepository FR-FCM-Z247 which is the validation cohort of an immunoprofiling study of women during pregnancy used in CytoNorm (*Van Gassen et al., 2019*). The samples we analyse come from the blood of one healthy donor which was replicated across 10 separately barcoded plates (i.e. batches). Each batch contained unstimulated cells and cells stimulated with both Interferon $\alpha$ and LPS. Each stimulated and unstimulated sample was duplicated (referred to as sample 1 and sample 2), giving four samples per plate. To assess the limitations of the three normalisation methods, we carried out two analyses. The first normalisation was only carried out on samples two using the stimulated samples two as replicate reference samples, while the second normalisation was done on samples 1 and 2 using both the stimulated and unstimulated samples one as reference samples. We performed CytoNorm as

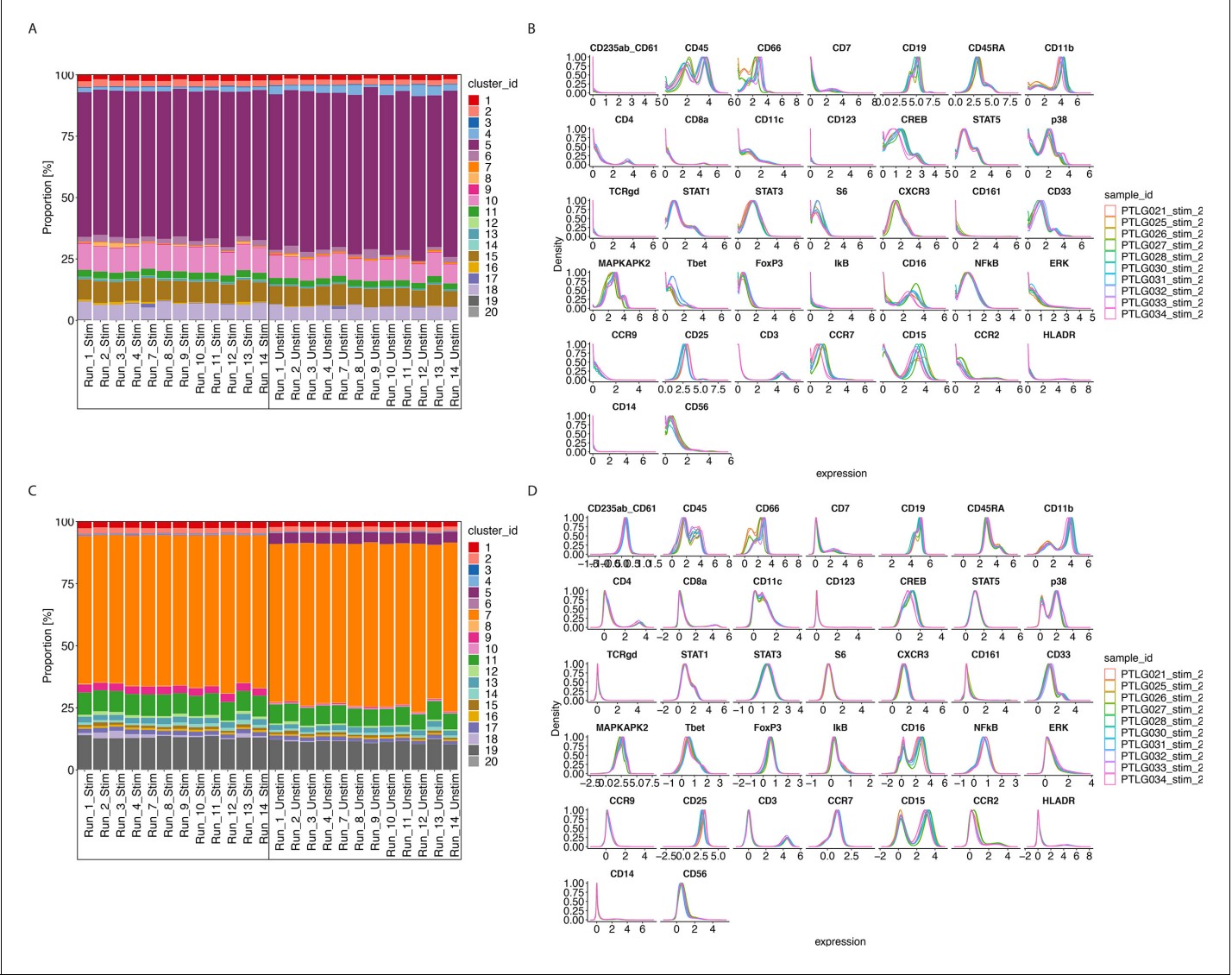

**Figure 6.** CytofRUV performance on two other datasets with multiple batches. (**A**) Barplot of proportions of clusters across 28 samples from the BatchAdjust dataset (*Schuyler et al., 2019*) before normalisation, by samples and coloured by cluster. Vertical black boxes contain the same sample (Stimulated or Unstimulated) replicated across 14 batches. (**B**) Protein expression distribution from the CytoNorm dataset (*Van Gassen et al., 2019*) before normalisation of all cells from the stimulated samples across 10 batches and coloured by batch. (**C**) Same as (**A**) but after CytofRUV normalisation with k = 10. (**D**) Same as (**B**) but after CytofRUV normalisation with k = 5.

The online version of this article includes the following figure supplement(s) for figure 6:

**Figure supplement 1.** Metrics to assess the effectiveness of the normalisation methods on the BatchAdjust dataset.

**Figure supplement 2.** Metrics to assess the effectiveness of the normalisation methods on samples two from the CytoNorm dataset using stimulated samples two as replicated reference samples.

**Figure supplement 3.** Metrics to assess the effectiveness of the normalisation methods on samples two from the CytoNorm dataset using both the stimulated and unstimulated samples one as replicated reference samples.

explained in *Van Gassen et al., 2019* and identified 25 clusters with FlowSOM and we used all clusters to define pseudo-replicates for CytofRUV.

For the first normalisation using only the stimulated samples as replicate reference samples, CytofRUV gives lower EMD for all samples, followed by CytoNorm which overall gives lower EMD when compared with BatchAdjust (*Figure 6—figure supplement 2A*, *Figure 6—figure supplement 2C*). CytoNorm corrects some differences in cluster proportions only in stimulated samples compared to the two other methods that give EMD generally similar to those in the raw data (*Figure 6—figure*

*supplement 2B*). However, according to the Hellinger distance and Silhouette scores with s$_{batch}$ higher to that in the raw data, the unstimulated samples still have batch effects after normalisation by all three methods (*Figure 6—figure supplement 2C*, *Figure 6—figure supplement 2E*). Similarly, the Silhouette scores of the stimulated samples indicate that none of the normalisation methods is able to successfully remove batch effects (*Figure 6—figure supplement 2C*).

In our second normalisation, when using both stimulated and unstimulated samples one as reference samples, lower Hellinger distances are found on both unstimulated and stimulated samples two for CytofRUV compared to BatchAdjust and CytoNorm, which have higher distances than those in the raw data for most samples. Better Silhouette scores are found for both samples for CytoNorm (*Figure 6—figure supplement 3B*, *Figure 6—figure supplement 3E*, *Figure 6—figure supplement 3C*) compared with the two other methods. Using our second class of diagnostic plots, we observe that CytofRUV with k = 5 effectively eliminates batch differences from protein expression across the 10 batches (*Figure 6B*) compared to that in the raw data (*Figure 6D*). It also gives the lowest EMD compared to both CytoNorm and BatchAdjust (*Figure 6—figure supplement 3A*, *Figure 6—figure supplement 3D*).

## Discussion

We have developed a new computational approach for analysing multiple CyTOF batches implemented in the CytofRUV R package. We showed how CytofRUV can reduce batch effects in three different datasets across multiple batches. Our method allows pooling of data from a large number of patients and conditions run across multiple batches, thereby enabling the integration of multiple CyTOF datasets.

Our approach adapts the RUV-III procedure to CyTOF by exploiting pseudo-replicates. We began by examining the batch effects found in CyTOF by comparing data from samples replicated across batches. To do this, we build an R-Shiny interface that highlights the presence of any batch effects in replicated samples using four different diagnostic plots: Median Protein Expression, Protein Expression Distributions, Clustering Results and Cluster Proportions. Finally, we compare CytofRUV with the two recently developed methods, BatchAdjust and CytoNorm, using three different datasets. Our results suggest that not only does CytofRUV frequently do better at removing unwanted variation in measured protein expression, making the distributions of these quantities more uniform across batches, but it also enhances the detection of biologically important differences embodied in the data across batches.

Using replicated samples to assess intra-site reproducibility and differences due to technical variation, we first showed how batch differences in protein expression in CyTOF datasets consisting of several CyTOF batches are comparable to biologically relevant differences within a single batch. In that context, confidently detecting of the impact of diverse treatments on distinct CLL cellular phenotypes would not be possible without correcting for such batch effects.

We developed an interactive R-Shiny application that exhibits batch effects in CyTOF studies. Different diagnostic plots can be selected by the user to display any batch effects on CyTOF data before and after normalisation.

We showed that not only are batch effects found in individual protein expression values which was also found in *Schuyler et al., 2019*, but also that batch differences affect samples differently. We observed the impact of batch differences on cell subtypes identification and how protein distributions are affected differently within cluster, something that was also found by *Van Gassen et al., 2019*. This suggests that it will not be sufficient to apply a single batch adjustment to the measurements on all cells, as is performed in BatchAdjust (*Schuyler et al., 2019*). We also noticed changes in the cluster abundances across batches.

RUV-III has been successfully applied to the Nanostring nCounter gene expression platform (*Molania et al., 2019*) and to single-cell RNA-seq data (*Lin et al., 2019*). Here, we adapted RUV-III to CyTOF data using as pseudo-replicates the cells of clusters in the different batches, and taking the collection of all protein expression values as 'negative controls'. We refer to the section 'Selection of negative control genes' (*Molania et al., 2019*) for this last point. Here, as there and in other contexts, before/after normalisation comparisons indicate that this can be effective. Our method is adaptable and flexible in that it allows the user to select different normalisations: it can be implemented for one or several replicated reference samples, it can also normalise specific clusters or all

clusters, and the dimension (k) of the unwanted variation can vary. In all cases, the user can visualise the diagnostic plots to assess the effectiveness of their normalisation. An important step of our method is the identification of biological subpopulations prior to normalisation, as this guides the selection of some or all clusters to define pseudo-replicates across batches.

To assess the abilities of CytofRUV, BatchAdjust and CytoNorm to remove batch effects, we computed three statistical metrics. We first compared the distributions of proteins across batches for replicated samples using EMD. Second, to assess how well each method corrects for differences in cluster proportions across batches, we computed the Hellinger distance between the paired cluster proportions of all replicated samples. Finally, to assess whether each method not only removes batch effects but preserves the biology, we computed Silhouette scores that quantify the cell-to-cell variation within cluster in relation to that between clusters.

Overall, CytofRUV had the best capability to make protein expression distributions more similar across batches. When compared with BatchAdjust and CytoNorm on different datasets it has the lowest EMD for all proteins for most samples in all three datasets and all four analyses performed, followed by CytoNorm. We further saw that batch differences are reduced within clusters by computing EMD by cluster for all proteins. We also saw CytoNorm's ability to correct for differences in cluster proportions as well as CytofRUV, both having the lowest Hellinger distances in two of the four analyses performed. Silhouette scores also indicate that in most of our analyses (two out of the three datasets) CytofRUV not only removes the batch effects but also improves biological accuracy of the clusters compared to the two other methods. Likewise, CytoNorm is performing well according to Silhouette scores in the second analysis of the third dataset. We also conclude that BatchAdjust has the least ability to remove batch effects in the three datasets we have explored here, according to those metrics. This conclusion might be a result the single batch adjustment performed by BatchAdjust to the measurements on all cells, whereas CytoNorm and CytofRUV both take into account the impact of batch differences at the cluster level.

We envisage that in some cases removing batch effects might lead to removal of clusters or to the identification of new biologically relevant clusters. Some clusters could be artefacts of the batch effects. We use $s_{biology}$ to determine the tightness and separation of the clustering, as it provides an evaluation of the clustering validity.

We observed that the B cell cluster (*Figure 3A*, cluster 16) is only present in the batch 1 of the CLL1 sample, and CytofRUV normalisation leads to the presence of this cluster (*Figure 3C*, cluster 2) in both batches. In practise, the user should examine and identify any new clusters post-normalisation, as some clusters might not be conserved after normalisation, and relate those to the clusters that were selected as pseudo-replicates. We also recommend the exploration of different clustering resolutions and how they change as the number of clusters increases, using clustering tree visualisation (*Zappia and Oshlack, 2018*) to relate the clusters pre and post-normalisation.

To further quantify the extend of batch effects in CyTOF datasets, it might be interesting to perform differential analyses of replicates across batches. Recent methods like diffcyt (*Weber et al., 2019*) have been developed that also uses FlowSOM for clustering to define cell populations, and empirical Bayes moderated tests adapted from transcriptomics for differential analysis. This method could be adapted to test for differential abundances of cell subpopulations and differential expression of proteins in the replicated samples across batches. In our first dataset containing two batches, differential analysis of protein expression is not suitable as we have too few replicates to permit a useful statistical test. However, we computed the differences in median expression for all proteins across batches before compared to that after CytofRUV normalisation for the CLL samples (ΔΔ, *Figure 3—figure supplement 4*.A) and the HC samples (ΔΔ, *Figure 3—figure supplement 4.B*). We observed that ΔΔ > 0 for most proteins especially for the CLL samples indicating reduction in batch differences. We did perform a differential analysis of the cluster abundances across batches using diffcyt and compare the results before and after normalisation. We also observed a decrease in the number of clusters that were found to have a significant difference in abundance across batches before normalisation (13 out of the 20 clusters) compared to that in the data after CytofNorm (7 out of 20 clusters).

One limitation of our method is that CytofRUV does rely on the assumption that at least one cell subpopulation is shared across the batches that would be used as our pseudo-replicates.

In experiments where large batch effects occur and no cell subpopulation is shared between batches, our method would not be applicable.

In this study, we considered three different datasets containing up to 24 samples and up to 12 batches. RUV methods have already been applied to hundreds and thousands of samples with Nanostring technology (*Molania et al., 2019*) and hundreds of samples with RNAseq. CytofRUV can also handle larger studies, and no memory limitation has been reached so far on other datasets containing hundreds of samples across several batches with up to 43 milllion cells. We ran the current analysis on a rstudio-server version 1.3.959–1 Professional for CentOS 6, taking about an hour per dataset, and we also tested it on other rstudio versions. Future work will involve collecting data from larger studies with more than dozens of batches to conduct further testing of memory limitations.

A requirement of the current CytofRUV method is the availability of enough material to have replicated reference samples in each batch. As previously mentioned for CytoNorm, and as we also observed for CytofRUV, in order to remove batch effects from all samples, it might be necessary to include more than one set of replicated references samples in the batch, in particular including samples that are similar to each of the main types of samples. In future work, we plan to explore estimating the unwanted variation using different replicated reference samples, each present in only some of the batches, to avoid the need for replicated reference samples in every batch. For example, the use of a carefully designed 'bridging' set of replicated reference samples analogous to long term reference samples in metabolomics (*Broadhurst et al., 2018*) should lead to the normalisation of large studies that is just as effective as that achieved using the same replicated reference samples throughout.

In summary, we proposed here a computational algorithm called CytofRUV that effectively enables batch effect reduction in mass cytometry with an adaptable normalisation method that detects heterogeneity of cellular responses across large-scale studies with multiple patients, cell types and conditions (e.g. treatment).

# Materials and methods

## CytofRUV

To remove the unwanted variation across multiple datasets and batches, we used fastRUV-III as previously described (*Lin et al., 2019*; *Molania et al., 2019*). Briefly, the data are first standardised before being fitted to the RUV-III model underlying all RUV analyses:

$$Y_{ij} = X_i \beta_j + W_i \alpha_j + \varepsilon_{ij}$$

Here, $Y_{ij}$ is the standardised expression value (arcsinh(intensity/5)) for protein $j$ in a cell $i$ with $i = 1, ldots, m$ and $j = 1, \ldots, n$, of $m$ cells and $n$ proteins. The standardisation is to have zero mean and unit standard deviation across all cells for all protein measurements.

$X_i$N represents the factors of interest for the sample giving cell $i$. $W_i$N represents the unwanted factors for that cell. The dimension of the unwanted factors is being denoted by $k$, $\alpha_j$N represents the coefficient of $W_i$N for protein $j$ in a cell $i$ and $\varepsilon_{ij}$ N is noise, typically $\varepsilon_{ij} \sim$ NN$(0, \sigma_j^2)$.

The data are normalised in six steps. First, all the data are clustered, typically using FlowSOM, although other clustering methods can be used. Second, group of cells are defined to be pseudo-replicates if they belong to the same subpopulation (i.e. cluster) but are in different batches, which can be done either on specific clusters or on all clusters. Third, cell-level residuals are computed by averaging all the protein measurements across those pseudo-replicates, and subtracting these averages from each measurement on each cell. In essence, differences of measurements on pseudo-replicates are considered as unwanted variation. The quantities $\alpha_j$ are then estimated from the singular value decomposition (SVD) of the $m \times n$ matrix of these residuals. Next the $k$-vectors of $W_i$ for the cells are then estimated using *all* proteins as 'negative controls'. Finally, the estimates of $W_i$ and $\alpha_j$ are multiplied to get an estimate of $W_i\alpha_j$, which is then subtracted from the $Y_{ij}$ to get the final adjusted data. Full details and a discussion of key issues can be found in *Lin et al., 2019* and *Molania et al., 2019*.

## BatchAdjust

For all the datasets, BatchAdjust was performed using the R code and usage instructions as described in *Schuyler et al., 2019*. We used the option of scaling the 95th percentiles with no transformation applied to the data before adjustment.

## CytoNorm

CytoNorm was performed as described in *Van Gassen et al., 2019* following the two steps provided in the R library CytoNorm to normalise the data.

## Earth Movers Distance

To quantify the similarity of protein expression distributions across batches, we computed the Earth Movers Distance, also called the Wasserstein metric, as described in *Van Gassen et al., 2019*. Briefly, the data are binned and we computed the pairwise EMDs across batches for the distribution of every protein over all cells as well as over the cells in each cluster. This was done for both the original dataset and the normalised datasets.

## Hellinger distance

To quantify the differences between cluster proportions across batches, we computed the Hellinger distances between the proportions found in samples replicated across different batches. This distance is defined for two probability $p = (p_i)$ and $q = (q_i)$ by $H(p, q)$, where

$$H(p, q)^2 = \tfrac{1}{2} \sum_{i=1}^{n} \left( \sqrt{p_i} - \sqrt{q_i} \right)^2.$$

We computed these distances for both the original and the normalised datasets.

## Silhouette scores

To assess the extent to which the data are grouped based on the batch effects as opposed to biological signals, we computed batch and biology Silhouette scores. Given a partitioning of all cells into groups, if $a_i$ denote the average Euclidean distance of the protein expression between the cell $i$ and all other cells in the group to which cell $i$ is assigned, and $b_i$ is the minimum of the average distance between the cell $i$ and any cells in other groups not containing cell $i$, then the silhouette coefficient of cell $i$ is calculated as

$$s(i) = \frac{b_i - a_i}{\max\{b_i, a_i\}}$$

The average of the silhouette values across cells using a particular grouping is called the silhouette score for that grouping. Silhouette score ranges from $-1$ to $+$one where positive values ($b_i$ is high and $a_i$ is low) indicate that cells are well matched to their own group. In this way, we computed the silhouette score $s_{batch}$ based on the batches as groups and the silhouette score $s_{biology}$ based on the grouping of the cells by subpopulation (i.e. clusters).

$s_{biology}$ is used to quantify the cell-to-cell variation within cell subpopulations compared to other subpopulations. Negative values mean that the data might be mis-clustered as it is more similar to a neighbouring cluster. For example, if two different biological relevant clusters would be merged into a single cluster, $s_{biology}$ will reflect this merging with a lower $s_{biology}$ value.

## Differential analysis

To perform a differential analysis of the cluster abundances across batches, we used diffcyt with the default method 'diffcyt-DA-edgeR' and default parameters with batch as the factor of interest for the differential tests. We performed this analysis both on the data before and after normalisation and calculated the number of significant detected clusters at 10% false discovery rate as described in the diffcyt workflow.

As the differential expression of proteins could not be performed with too few replicates, as with a t-test, we computed the medians $M_{m,b}$ for each marker $m$ and each batch $b$ across the 12 paired samples in a given cluster. We then computed the difference of medians between batches $\Delta = M_{m,2} - M_{m,1}$ across the 12 paired samples. We computed those differences *raw* on the data

before and $\Delta norm$ after normalisation and then plotted the difference of those values, that is $\Delta\Delta = \Delta raw - \Delta norm$ in *Figure 3—figure supplement 4*.

## R package CytofRUV

Our algorithm is implemented in the R package 'CytofRUV' and is available at: (www.github.com/mtrussart/CytofRUV) (*Trussart, 2020*). Installation and R code usage instructions for both the R package and the R-Shiny application can be found on the GitHub page. Users are required to provide: the FCS files from all samples in the study, a metadata file containing the details of each sample and a panel file containing the details of all proteins used in the study. The R-Shiny application allows the user to explore the data and identify batch effects across replicates using the diagnostic plots previously mentioned: Median Protein Expression, Protein Expression Distributions, Clustering Results and Cluster Proportions using samples replicated across batches. It can be explored on all the data or on a subsample of the data. The normalize_data function allow the user to adjust for batch effects with parameter settings for the CytofRUV algorithm, such as which replicated samples to use and the value of k. The pipeline and scripts used to generate the results described in this manuscript is also available in the supplementary data.

## PBMC samples from patient and healthy donor

Blood was obtained from healthy donors (via the Victorian Donor Blood Registry) and patients with CLL (via the Royal Melbourne Hospital, Australia). All patients consented under Melbourne Health HREC 2016.305 and samples were analysed under HREC 2016.066. PBMCs were isolated using standard Ficoll-based methods and cryopreserved.

## Mass cytometry

Cells were thawed and stained for viability with cisplatin. Cells were then fixed with paraformaldehyde (PFA: Electron Microscopy Sciences, Hatfield, PA, USA) to a final concentration of 1.6% for 10 min at room temperature. Cells were pelleted and washed once with cell staining medium (CSM, PBS with 0.5% BSA and 0.02% sodium azide) to remove residual PFA and stored at $-80°C$.

Cells were barcoded using 20-plex palladium barcoding according to manufacturer's instructions (Fluidigm, South San Francisco, CA, USA). Following barcoding, cells were pelleted and washed once with cell staining medium (PBS with 0.5% BSA and 0.02% sodium azide) to remove residual PFA. Cells were stained with CD16/CD32 for 10 min and surface antibodies (*Table 2*) for 30 min at room temperature. Cells were permeabilized with 4°C methanol for 10 min. Cells were washed three times with CSM and stained with intracellular antibodies (*Table 2*) for 30 min at room temperature. Cells were washed with CSM, then stained with 125 nm $^{191}$Ir/$^{193}$Ir DNA intercalator (Fluidigm, South San Francisco, CA, USA) in PBS with 1.6% PFA at 4°C overnight. Cells were washed once with CSM, washed three times with double-distilled water and filtered to remove aggregates and resuspended with EQ normalisation beads immediately before analysis using a Helios mass cytometer (Fluidigm, South San Francisco, CA, USA). Throughout the analysis, cells were maintained at 4°C and introduced at a constant rate of ~300 cells/sec.

## Processing data for mass cytometry

Data concatenation, normalisation and debarcoding are done using the R Catalyst package merging the two batches (RUV_1B and RUV3B) when applying the Finck normalisation. The R script used to generate this preprocessing (CytofRUV_preprocessing_dataset.R) is also available in the supplementary data.

## Flow repository

The FCS files from this study are available at flow repository ID FR-FCM-Z2L2.

## Acknowledgements

MT and TPS were supported by NHMRC Program Grant 1054618. DHDG is supported by a NHMRC Senior Research Fellowship (1158024), a Cancer Council of Victoria Grant-in-aid (1146518) and

support from the Victorian Comprehensive Cancer Centre. CET is supported by an Australian NHMRC Early Career Fellowship (1089072).

We are grateful to Dr MaryAnn Anderson and Prof Andrew Roberts for the provision of patient material.

This work was made possible through Victorian State Government Operational Infrastructure Support and Australian Government NHMRC IRIISS. The authors thank Dr Andrew Mitchell (University of Melbourne) for assistance with mass cytometry maintenance/operation. This work was performed in part at the Materials Characterisation and Fabrication Platform (MCFP) at the University of Melbourne and the Victorian Node of the Australian National Fabrication Facility (ANFF).

We thank Prof. Damien Hicks, Dr Chris Woodruff and Dr Ramyar Molina for their suggestions during this study. We thank Dr Anna Quaglieri for her suggestions concerning the CytofRUV R package. We also thank Prof. Ronald P Schuyler, Prof. Debashis Ghosh, Prof. Elena Hsieh, Ms Helena Crowell, Prof. Jean Yang, Yingxin Lin and Dr Greg Finak as well as the reviewers for their helpful comments on the manuscript.

## Additional information

### Funding

| Funder | Grant reference number | Author |
|---|---|---|
| National Health and Medical Research Council | 1054618 | Marie Trussart<br>Terence P Speed |
| National Health and Medical Research Council | 1158024 | Daniel HD Gray |
| Cancer Council Victoria | 1146518 | Tania Tan<br>Daniel HD Gray |
| National Health and Medical Research Council | 1089072 | Charis E Teh |
| Perpetual Impact Philanthropy | IPAP2019/1437 | Charis E Teh |
| UROP Fellowship | | Lawrence Leong |

The funders had no role in study design, data collection and interpretation, or the decision to submit the work for publication.

### Author contributions

Marie Trussart, Conceptualization, Data curation, Software, Formal analysis, Supervision, Validation, Visualization, Methodology, Writing - original draft, Writing - review and editing; Charis E Teh, Conceptualization, Resources, Formal analysis, Supervision, Funding acquisition, Validation, Investigation, Methodology, Project administration, Writing - review and editing; Tania Tan, Investigation; Lawrence Leong, Data curation, Software; Daniel HD Gray, Conceptualization, Resources, Supervision, Funding acquisition, Methodology, Project administration, Writing - review and editing; Terence P Speed, Conceptualization, Resources, Supervision, Funding acquisition, Validation, Methodology, Project administration, Writing - review and editing

### Author ORCIDs

Marie Trussart https://orcid.org/0000-0002-7258-7272
Charis E Teh https://orcid.org/0000-0002-9745-2876
Daniel HD Gray http://orcid.org/0000-0002-8457-8242
Terence P Speed https://orcid.org/0000-0002-5403-7998

### Ethics

Human subjects: All patients provided written informed consent and the study was approved by Human Research Ethics Committees/Institutional Review Boards: RMH (2005.008, 2012.244, 2016.305,2016.066) and the Walter and Eliza Hall Institute (G15/05).

Decision letter and Author response
Decision letter https://doi.org/10.7554/eLife.59630.sa1
Author response https://doi.org/10.7554/eLife.59630.sa2

## Additional files

### Supplementary files
• Supplementary file 1. Scripts to generate the CytofRUV figures from the dataset FR-FCM-Z2L2. CytofRUV_preprocessing_dataset.R is the script that was used to generate the preprocessing on the dataset: data concatenation, normalisation and debarcoding are done using the R Catalyst package merging the two batches (RUV_1B and RUV3B) when applying the Finck normalisation. CytofRUV_-Figures.R was used to generate the figures described in this manuscript and uses the CytofRUV R package with the Metadata.xlsx and Panel.xlsx files. The package is available at: www.github.com/mtrussart/CytofRUV. Installation and R code usage instructions for both the R package and the R-Shiny application can be found on the GitHub page.

• Transparent reporting form

### Data availability
The fcs files from this study are available at Flow Repository, ID FR-FCM-Z2L2.

The following dataset was generated:

| Author(s) | Year | Dataset title | Dataset URL | Database and Identifier |
|---|---|---|---|---|
| Trussart M, Teh CE, Tan T, Leong L, Gray DHD, Speed TP | 2020 | CytofRUV dataset | http://flowrepository.org/id/FR-FCM-Z2L2 | FlowRepository, FR-FCM-Z2L2 |

The following previously published dataset was used:

| Author(s) | Year | Dataset title | Dataset URL | Database and Identifier |
|---|---|---|---|---|
| Gassen S | 2019 | Immune Clock of Pregnancy Validation - Controls | http://flowrepository.org/id/FR-FCM-Z247 | FlowRepository, FR-FCM-Z247 |

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
