## [Decision Letter]

**Acceptance summary:**

The work is an important contribution to the field of cytometry. It provides an objective and well thought out normalization procedure for mass cytometry and potentially even fluorescence flow cytometry.

**Decision letter after peer review:**

Thank you for submitting your article "CytofRUV: Removing unwanted variation to integrate multiple CyTOF datasets" for consideration by *eLife*. Your article has been reviewed by four peer reviewers, one of whom is a member of our Board of Reviewing Editors, and the evaluation has been overseen by Detlef Weigel as the Senior Editor. The following individuals involved in review of your submission have agreed to reveal their identity: Nima Aghaeepour (Reviewer #1); Anna Belkina (Reviewer #3); Sofie Van Gassen (Reviewer #4).

The reviewers have discussed the reviews with one another and the Reviewing Editor has drafted this decision to help you prepare a revised submission.

Summary:

The authors present a new Cytof normalization approach based on RUV III that has proven useful for other technologies including RNASeq, single-cell RNAseq and nanostring. The reviewers all agreed that this was a strong manuscript that makes an important contribution to an area of the field that remains under-served, and they unanimously recommended publication.

Essential revisions:

Several concerns arose during review that the authors should address to strengthen the results and improve the presentation.

1) Given the size of typical cytometry data sets (millions of cells and hundreds of samples), how does this approach scale? What are the limitations? The Discussion mentions large-scale studies, but how large in practice?

2) Most importantly, reviewers raised questions around the evaluation of the normalization procedures. Specifically, since the data are re-clustered after normalization, and performance of normalization was assessed against the re-clustered data (and while the reviewers agreed this made sense), they were concerned about how a negative impact of normalization could be assessed. Specifically, if normalization failed, leading to fusion of biologically relevant clusters post-normalization, how could that be detected? It was not clear that this would be captured by S_biology_ (as pre- and post-normalization S_biology_ measures are not compared), nor how and if the proposed figures and evaluation measures could be interpreted to detect this situation. The reviewers felt this aspect needed further exploration and discussion.

3) Finally the reviewers were looking for more guidance from the presentation around the relative ranking of the different methods. The authors should more clearly present their conclusions about the relative performance of the methods and state which methods are performing well or poorly and why, and if the results are not conclusive, this should be stated and explained.

---

## [Author Response]

Essential revisions:Several concerns arose during review that the authors should address to strengthen the results and improve the presentation.1) Given the size of typical cytometry data sets (millions of cells and hundreds of samples), how does this approach scale? What are the limitations? The Discussion mentions large-scale studies, but how large in practice?

We thank the reviewers for their question and we updated the Discussion of the manuscript with more details regarding large-scale studies and their limitations in the revised manuscript.

“In this study we considered three different datasets containing up to 24 samples and up to 12 batches. […] For example, the use of a carefully designed “bridging” set of replicated reference samples analogous to long term reference samples in metabolomics (Broadhurst et al., 2018) should lead to the normalization of large studies that is just as effective as that achieved using the same replicated reference samples throughout.”

2) Most importantly, reviewers raised questions around the evaluation of the normalization procedures. Specifically, since the data are re-clustered after normalization, and performance of normalization was assessed against the re-clustered data (and while the reviewers agreed this made sense), they were concerned about how a negative impact of normalization could be assessed. Specifically, if normalization failed, leading to fusion of biologically relevant clusters post-normalization, how could that be detected? It was not clear that this would be captured by S_biology_ (as pre- and post-normalization S_biology_ measures are not compared), nor how and if the proposed figures and evaluation measures could be interpreted to detect this situation. The reviewers felt this aspect needed further exploration and discussion.

The reviewers are raising an important aspect to take into account in the evaluation of the normalization. We acknowledge that in some cases there might be loss of clusters or the identification of new biologically relevant clusters. An important aspect to take into account in this discussion is that the same number of clusters is conserved before and after normalization. To assess the performance of the normalization, we clarify out use of S_biology_ and provide an example of how users can validate their clusters. We expanded the exploration and discussions of the questions raised here in the Discussion in the revised manuscript.

3) Finally the reviewers were looking for more guidance from the presentation around the relative ranking of the different methods. The authors should more clearly present their conclusions about the relative performance of the methods and state which methods are performing well or poorly and why, and if the results are not conclusive, this should be stated and explained.

We added more descriptions in the relative performance of each method for all three datasets (Results) and clearly stated the conclusions from those analysis in the Discussion in the revised manuscript.